# A Mathematical Model Combined with Radar Data for Bell-Less Charging of a Blast Furnace

**Meng Li** [1], **Han Wei** [1], **Yao Ge** [1], **Guocai Xiao** [2] **and Yaowei Yu** [1,*]

1   State Key Laboratory of Advanced Special Steel, Shanghai Key Laboratory of Advanced Ferrometallurgy, School of Materials Science and Engineering, Shanghai University, Shanghai 200444, China; L_limeng@outlook.com (M.L.); weihan@shu.edu.cn (H.W.); ge_geyao@163.com (Y.G.)
2   Jinheng Information Technology Company, Jiangsu Province, Nanjing 210035, China; xiaoguocai@yeah.net
*   Correspondence: yaowei.yu@hotmail.com

**Abstract:** Charging directly affects the burden distribution of a blast furnace, which determines the gas distribution in the shaft of the furnace. Adjusting the charging can improve the distribution of the gas flow, increase the gas utilization efficiency of the furnace, reduce energy consumption, and prolong the life of the blast furnace. In this paper, a mathematical model of blast furnace charging was developed and applied on a steel plant in China, which includes the display of the burden profile, burden layers, descent speed of the layers, and ore/coke ratio. Furthermore, the mathematical model is developed to combine the radar data of the burden profile. The above model is currently used in Nanjing Steel as a reference for operators to adjust the charging. The model is being tested with a radar system on the blast furnace.

**Keywords:** blast furnace; charging system; mathematical model; radar data; burden distribution

---

## 1. Introduction

The raw material used in the production of a blast furnace is called burden. It is mainly composed of coke, sinter, and pellet. The prepared burden is loaded into a hopper. After a series of transportation steps on the top of the furnace, it falls onto a rotating chute, when the exit of the hopper opens. Then, the material enters the blast furnace and forms burden layers in the throat. At the bottom, hot air is blown into the furnace to burn the coke, producing carbon monoxide and hydrogen that act as reductants. The rising reductants react chemically with iron oxide in burden and the iron oxide becomes hot metal after the reduction and melting. In the reduction, impurities in iron ore combine with the added flux to form molten slag. Hot metal and slag are discharged from tapholes at the bottom of the furnace. After treatment, molten slag is used as a raw material for cement and hot metal is transported to basic oxygen furnace by torpedo.

Some studies have shown that the charging affects the chemical reaction between the layers and reducing gas in the shaft of the furnace. The layers influence the shape of the cohesive zone and energy utilization efficiency of the furnace [1]. Therefore, the charging system is of great significance for the operation of the blast furnace. Due to the high temperature, high pressure, and dusty environment of the blast furnace, the internal state of the furnace cannot be measured. Even though furnace top temperature detection equipment, cross temperature measurement, and furnace top infrared cameras are already applied in the furnace, they still cannot accurately provide information about the burden layers in the furnace. Simulation technology is a powerful method combining computer, mathematics, physical engineering, and chemical engineering. It is helpful to observe the phenomena that are difficult to be directly measured in practice. Meanwhile, it saves costs, time, and materials in experiments [2].

Many studies have used mathematical models to study the charging system of the blast furnace. Yoshimasa et al. [3] developed a simulation model for the burden distribution of blast furnace charging and studied the trajectory of the raw material, burden descent, and mixing layer, providing important information for the subsequent model development. Pohang Iron and Steel Company proposed a radial distribution function of the burden and applied it in an actual blast furnace to study the distribution characteristics of the burden and to improve the distribution of the gas flow [4]. Krishman et al. [5] developed a mathematical model for the optimization of bell-less charging, and the calculation results were consistent with the actual data. Saxén and Hinnelä [6] developed a bell-less burden distribution model on the basis radar measurement, and the dependence between the layer thickness and charging variables was modeled by neural networks [7]. Nag [8] proposed a mathematical model of the bell-less top to calculate the trajectory of the burden in charging. Park et al. [9] analyzed the blast furnace charging system by developing a burden descent model and a gas flow model, and compared the results with those from a 1/12-scaled model experiment. Samik et al. [10] proposed a general target methodology to estimate the stock profile in the blast furnace, where the burden distribution is based on experiments in different scaled models of a blast furnace with various materials. Shi et al. [11] proposed a new model of stockline profile formation in which equations were developed for the inner and the outer repose angles by considering the influence of the burden's vertical and horizontal flow.

The above mathematical models were developed based on some assumptions and different operating conditions. Therefore, it is quite difficult to apply them to get accurate results of the burden layer for other furnaces. If a charging model can be combined with a reliable burden surface detection method, the reliability of the calculated burden profile information can be increased. For example, rotating radar detection technology can more accurately measure the height of each point of the burden surface even under severe conditions, such as complete darkness and high-dust atmosphere, than a mechanical stock rod [12]. Therefore, in a black-box environment, vibration, and strong airflow, burden distribution online measurement should be stable and accurate. Considering the limits of the radar method, radar data also includes noise and can only reflect the surface shape of the object. Through the combination of radar, data processing, and the charging mathematical model, the error and noise of radar data can be reduced a lot. Furthermore, the shape of the burden surface and the structure of the burden layers can be better estimated [13].

Some scholars have studied the application of radar in blast furnaces, and some achievements have been made. Liu et al. [14] used radar data containing information of the burden surface situation and cross thermometer data reflecting the trend change of the burden surface with time as the training data for a fuzzy neural network to classify and predict the burden surface. Gao et al. [15] suggested that visualization and simulation is a new technology to monitor the charging system and to help the operation of blast furnaces. Based on the real multi-radar data, Zhu et al. [16] estimated the burden profile by a cubic-curve equation at the end of a multi-loop charging. Furthermore, the burden profile before the next multi-loop charging was calculated by considering the impact of the burden descent. Li et al. [17] used fuzzy c-means clustering to classify a large amount of burden surface radar data and proposed a multiple-model set of the burden surface. The real-time burden surface data were matched with the model to produce the expected burden surface. A reconstruction algorithm based on phased array radar data was proposed by Zhang [13] to extract the data of the blast furnace charging line and it was shown to have high efficiency and high accuracy. Tian et al. [18] developed a radar detection-based model for the prediction of the burden surface shape to develop a charging strategy and the results showed that the proposed model had the advantages of higher prediction accuracy for both local details and global shape than mechanical stock rods. In another publication [19], they proposed an innovative data-driven model for predicting the distribution of the burden descent speed. This model has the ability to better characterize the variability in the radial distribution of the burden descent speed than a pure mathematical model based on Newton's second law. Miao et al. [20] proposed a new calculation method of shape fusion of the material line based on a stacking method, which can directly calculate the shape of the material surface and improve the measurement accuracy by 4.8%, compared to the

first principle mathematical model. Li et al. [21] recently presented a similar model to improve the measurement accuracy of the burden profile and to use this in modeling of the burden distribution.

From the above literature and statements, there are no publications presenting a combination of radar data and a mathematical model to test, modify, and improve the model. Therefore, this paper will concentrate on this issue.

## 2. Mathematical Model and Radar Data Treatment

### 2.1. Mathematical Model Structure

Locations of raw material trajectories, shape of the burden profile, and ratio of ore to coke on the top of the blast furnace are gained by calculating the charging process. This is important for predicting the reducing gas distribution and chemical reactions between layers and the gas in the shaft of the blast furnace. Therefore, combining the mathematical model of the charging with the experience of the production and radar data, the operation of the blast furnace can be optimized to become more stable and efficient.

From the raw material in the hopper to the formation of burden layers in the throat of the furnace, the charging process is decomposed into the trajectory of the burden flow, burden profile, burden layer structure, and burden distribution evaluation. The four steps are calculated by four models: Burden flow trajectory model, burden profile model, burden distribution model, and burden evaluation model, and are shown in Figure 1.

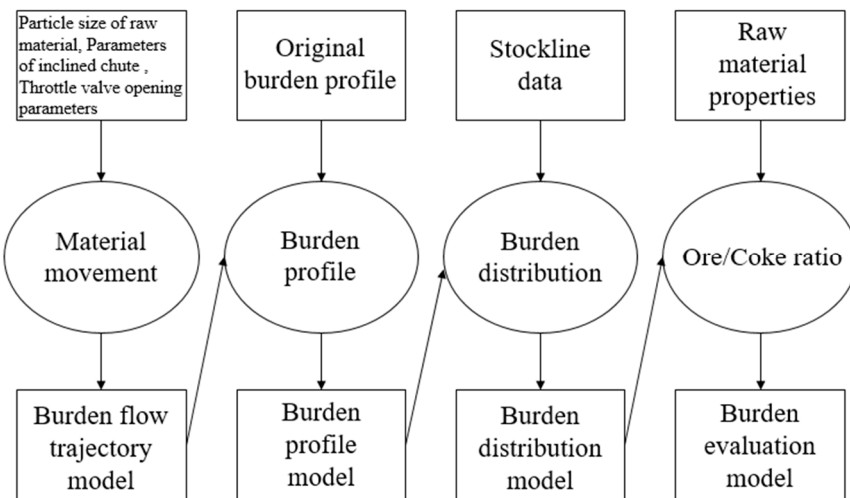

**Figure 1.** Mathematical model structure.

In Figure 1, after the calculation of the material movement by Newton's second law, the velocity of the burden at the hopper exit, velocity of the burden into the chute, and velocity of the burden leaving the chute tip are obtained successively. Then, based on the inclination angle and rotation speed of the chute, the trajectories of the raw material after leaving the chute tip are calculated. From the trajectories and burden profile model, the coordinates of a new burden profile can be calculated based on the original one. Then, the descent speed of the burden is used to modify the new burden profile and the modified one is saved in the database. In the next calculation, the modified one is considered as an original profile and the procedure is repeated. Finally, the ratio of ore to coke in the last two profiles is obtained and is used as a criterion of evaluation.

#### 2.1.1. Burden Flow Trajectory Model

This part studies the movement of raw material particles and velocities of the material from the hopper to the trajectory of the burden flow. This part is divided into four sections and is shown in

Figure 2. It includes the velocity of the burden at the hopper exit, velocity of the burden into the chute, velocity of the burden leaving the chute tip, and trajectory of the burden after leaving the chute tip.

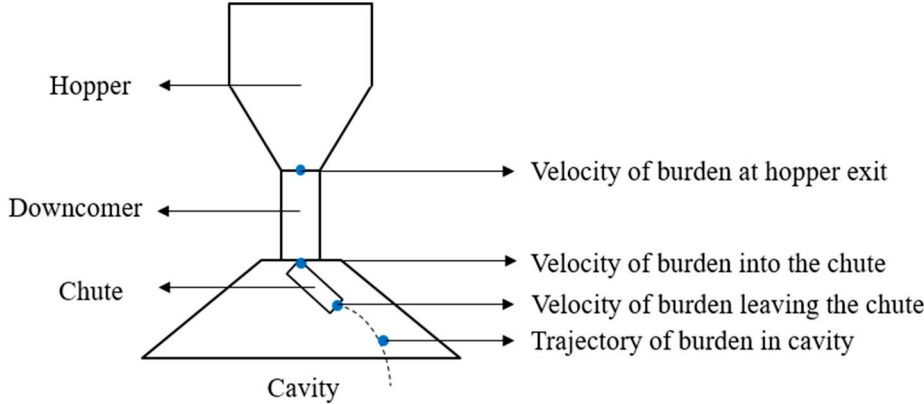

**Figure 2.** Trajectory of the burden flow passing through the bell-less top.

The raw material flows out from the exit of the hopper in a funnel form, and its velocity ($V_0$) can be described by the hydraulic formula [22]:

$$V_0 = Q/\pi(2S/C - d_i/2)^2, \tag{1}$$

where $S$, $C$, $d_i$, and $Q$ express the projection area of the throttle valve (m$^2$), circumference of the throttle (m), average particle size of the burden (m), and flow rate of the burden out the throttle valve (t/s), respectively.

According to the literature [23], the relationship between $Q$ and $A$ (throttle valve opening) is as follows:

Ore:

$$Q = 1 \times 10^{-5}A^3 - 7 \times 10^{-4}A^2 + 0.0366A - 0.5; \tag{2}$$

Coke:

$$Q = 3 \times 10^{-5}A^3 - 2.7 \times 10^{-3}A^2 + 0.103A - 1.29. \tag{3}$$

Before reaching the chute, raw material particles free fall with an initial velocity $V_0$ and collide with the wall of the downcomer (see Figure 2), causing a loss of energy (velocity), which can be calculated by the velocity attenuation factor $k$. Therefore, the velocity of the particles entering the chute is calculated by:

$$V_1 = \sqrt{k \cos \alpha (V_0^2 + 2g(h + b/\sin \alpha))}, \tag{4}$$

where $\alpha$, $h$, $b$, $g$, and $A$ define the inclination angle of the chute in the vertical direction (°) (see Figure 3), height of the downcomer (m), distance from the chute suspension point to the bottom of the chute (m), acceleration due to gravity (9.81m/s$^2$), and throttle valve opening of the hopper's exit (°), respectively.

The particles fall into the chute at the velocity ($V_1$) and are mainly subjected to gravitational force ($F_1$), supportive force ($F_2$), frictional force ($F_3$), and centrifugal force ($F_4$) as shown in Figure 3. The applied forces on particles along the chute can be expressed by:

$$F_1 = mg \tag{5}$$

$$F_2 = mg \sin \alpha - \omega^2 lm \sin \alpha \cos \alpha \tag{6}$$

$$F_3 = \mu F_2 \tag{7}$$

$$F_4 = \omega^2 ml \sin \alpha \tag{8}$$

$$\sum F = F_1 + F_2 + F_3 + F_4, \tag{9}$$

where $\omega$, $m$, $l$, and $\mu$ express the rotation speed of the chute (r·S$^{-1}$), mass of the burden (kg), length of the chute (m), and coefficient of dynamic friction (−), respectively.

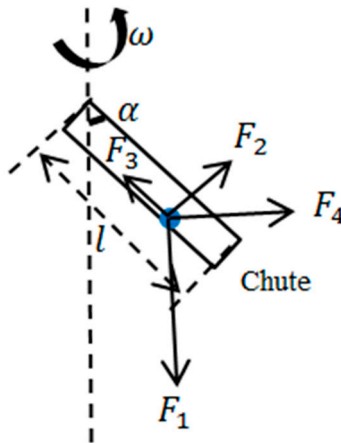

**Figure 3.** Schematic diagrams of the applied forces on the particle flow along the chute.

According to Newton's second law, the velocity $V_2$ of particles leaving the chute end can be calculated and it is decomposed into the horizontal velocity $V_h$, vertical velocity $V_v$, and tangential velocity $V_t$ as follows:

$$V_2 = \sqrt{\omega^2 \sin\alpha(\sin\alpha + \mu\cos\alpha)l^2 + 2g(\cos\alpha - \mu\sin\alpha)l + (V_1\cos\alpha)^2}, \tag{10}$$

$$V_h = V_2 \sin\alpha \tag{11}$$

$$V_v = V_2 \cos\alpha \tag{12}$$

$$V_t = r\omega \tag{13}$$

After leaving the chute end, the burden moves with the velocity of $V_2$ in the throat until it falls onto the burden surface. In the movement, burden particles are subjected to gravitational force, buoyancy force, and the drag force of gas. The influence of the latter two forces on the movement of the burden is very small and is ignored [22]. Therefore, the movement of the burden is treated as a slant throw movement with gravity, as is shown in Figure 4.

The slant throw movement of particles can be decomposed into two directions: The radius of the throat ($S_r$) and the tangential direction of the radius ($S_t$). Therefore, the distance of particles from the center line of the furnace ($S$) to the falling point of particles with the burden profile can be calculated by:

$$S_r = r + (V_2 \sin\alpha)t \tag{14}$$

$$S_t = \omega r t \tag{15}$$

$$S = \sqrt{S_r^2 + S_t^2} \tag{16}$$

where $r$ and $t$ are the radial distance from the chute tip to the center line of the blast furnace (m) and the movement time of particles between leaving the chute tip and reaching the burden profile (S).

When the burden moves below the zero value of the stock line, the vertical distance between material particles and the zero value of the stock line can be expressed by:

$$H = h_2 - (h_0 - h_1), \tag{17}$$

where $h_0$, $h_1$, $h_2$, and $H$ define the distance from the chute suspension point to the zero stock line (m), distance from the chute suspension point to the end of the chute (m), vertical distance of the material after leaving the chute tip (m), and distance between the material and zero value of the stock line (m), respectively.

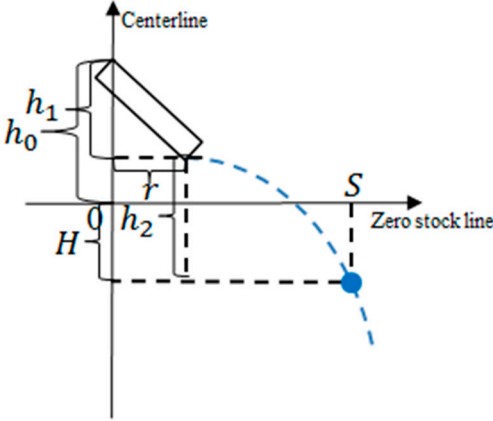

**Figure 4.** The trajectory of the material in the cavity, where $h_0$ is the distance from the chute suspension point to the zero value stock of the line (m), $h_1$ is the distance from the chute suspension point to the end of the chute (m), $h_2$ is the vertical distance of the material after leaving the chute tip (m), and $H$ is distance between the material and zero value of the stock line (m).

### 2.1.2. Burden Profile Model

After the raw material moves from the chute tip, it is uniformly dumped to the burden surface of the furnace and forms a new one. For multi-ring charging programs, the material forms a new shape of the burden surface with several concentric piles. There is a certain width of the burden mass flow, after the material leaves the chute tip. Therefore, we describe the mass flow by two trajectories: The main trajectory of the material flow (mass trajectory) and the lower material flow, as shown in Figure 5.

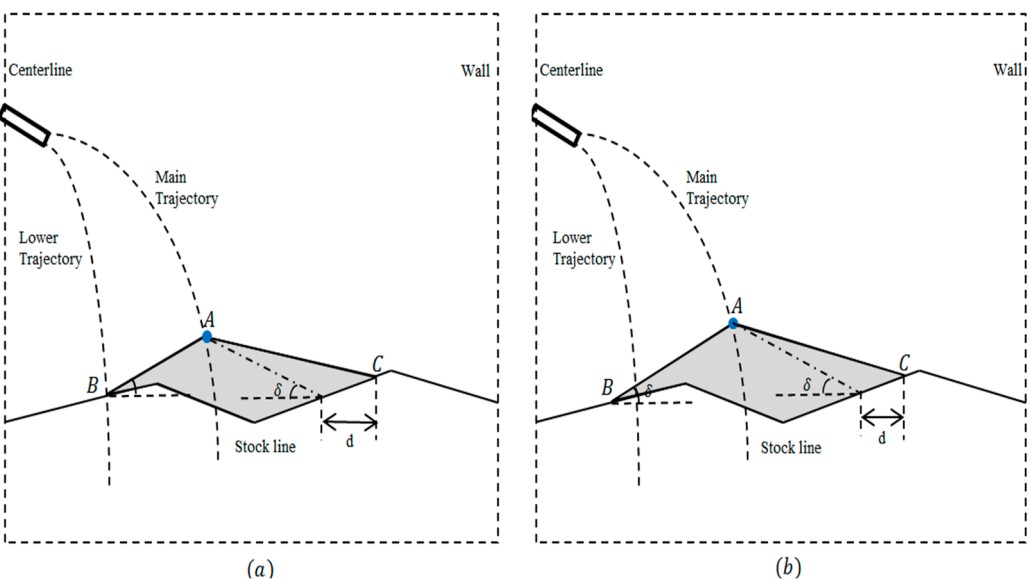

**Figure 5.** Schematic diagram of a new surface formation on an original one, where $\delta$ is the repose angle of the burden. (**a**) the location of the inner foot B is the intersection of the lower trajectory with the previous burden profile; (**b**) the material keeps the repose angle slipping on the inner surface of the profile.

Along the radial direction of the furnace, the surface of a new burden profile can be described by three points as shown in Figure 5: Inner foot B of the pile, outer foot C of the pile, and apex A of the pile.

When the material settles down on the inner surface of a new pile, the angle of the new pile is less than the repose angle of the raw material, and the location of the inner foot B is the intersection of the lower trajectory with the previous burden profile as shown in Figure 5a. When the material slips on the inner surface of the pile, the inner surface reaches the repose angle of the material and will keep the repose angle, as shown in Figure 5b. Therefore, more material moves towards the center of the furnace and the inner foot B is decided by the volume conservation of the raw material. For the apex A calculation, it is given by the intersection of the main trajectory with the previous burden profile. The right-side material of the main trajectory falls to the outer side of the apex A and forms the outer surface of the pile. Because the particles have a velocity component in the radial direction of the furnace, they roll along the outer surface of the pile for a while. The rolling distance of the material is constant for different materials and is 0.7 m for coke and 0.5 m for ore [23].

The new burden profile is described by the A, B, and C points along the radial direction as shown in Figure 6. It is divided into several regions for integration to obtain the volume. Therefore, the new burden volume is calculated as follows and is equal to the volume of the material in each batch:

$$V = 2\pi \int (f_{new}(r) - f_{ori}(r)) r dr = 2\pi \int_{n_i}^{n_{i+1}} (f_m(r) - f_n(r)) r dr, \tag{18}$$

where $r$, $f_{new}(r)$, $f_{ori}(r)$, $f_m(r)$, and $f_n(r)$ are the distance from the center line (m), new burden profile function, original burden profile function, mth burden profile function, and nth burden profile function, respectively.

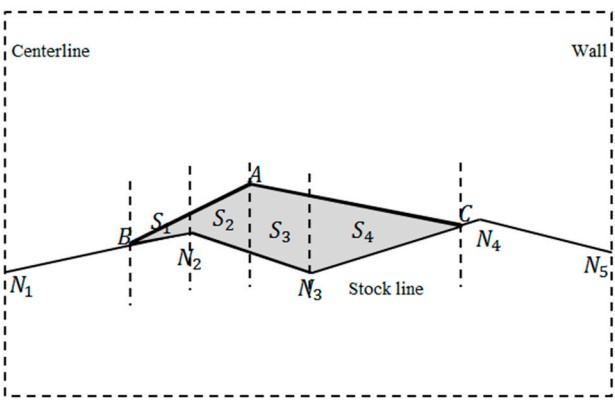

**Figure 6.** Burden volume partition integral.

### 2.1.3. Burden Distribution Model

Tapping of hot metal and molten slag from the taphole in the hearth as well as combustion and gasification of coke give the space for the burden to descend in the furnace. The shape of the layer distribution changes during the descent. Therefore, it is necessary to modify the burden profile according to the burden descent. In practice, two to four mechanical stock rods are used to measure the descent of the burden at fixed locations.

According to the measurement, Nishida et al. [24] proposed a velocity distribution of the burden descent along the radial direction of the furnace ($V_{(r)}$), which is measured directly by a profilometer and is a linear assumption based on the measured data. From the velocity distribution, the rate of the burden descent is slower in the center and faster near the wall of blast furnace:

$$V_{(r)} = a + br \tag{19}$$

$$a = \frac{R_1 V_{ch} - \frac{2}{3} R_0 V_{rod}}{R_1 - \frac{2}{3} R_0} \tag{20}$$

$$b = \frac{V_{rod} - V_{ch}}{R_1 - \frac{2}{3} R_0} \tag{21}$$

where $V_{(r)}$ is the velocity component of the burden descent at the radial position $r$ (mm/s), $R_0$ is the throat radius (m), $R_1$ is the distance of the profilometer from the center line (m), $V_{ch}$ is the average velocity of the burden descent (mm/s), and $V_{rod}$ is the descent velocity at the profilometer (mm/s).

The descent velocity of the burden is only vertical in the furnace throat ($V_r$), and divides into vertical ($V_h$) and horizontal ($V_r$) components in the shaft region as follows:

$$V_h = V_{(r)} \sin \beta, \; V_r = V_{(r)} \cos \beta, \tag{22}$$

where $\beta$ defines the shaft angle of the furnace (°).

When different kinds of raw materials are charged into the throat of the furnace, the structure of alternating layers (ore layer and coke layer) forms. The burden still keeps its layer structure in the descent. Therefore, the layer structure of the burden can be used to study the distribution of the ratio of ore to coke and the permeability of the burden in the shaft.

### 2.1.4. Burden Evaluation Model

In order to gain a reasonable material distribution and help the operation of the blast furnace, it is necessary to evaluate the distribution of the layers. The particle average size and strength of coke are much larger and higher than those of sinter. Coke remains in a solid state to 1500 °C while sinter softens and melts below the cohesive zone in the furnace. In the blast furnace, coke has better permeability than sinter. Therefore, the mass ratio of ore to coke is used to evaluate the burden of the coke state and is calculated as follows:

$$K_{O/C} = \frac{\Delta M_O}{\Delta M_C} = \frac{[f_O(r)_n - f_C(r)_n]\rho_O}{[f_C(r)_n - f_O(r)_{n-1}]\rho_C}, \tag{23}$$

where $\Delta M_O$ is the mass of the ore layer (kg), $\Delta M_C$ is the mass of the coke layer (kg), $f_O(r)_n$ is the function of ore in the nth layer, and $f_C(r)_n$ is the function of coke in the nth layer. $\rho_O$ and $\rho_C$ are the ore bulk density (kg/m$^3$) and coke bulk density (kg/m$^3$), respectively.

### 2.2. Radar Detection Measurement

Rotating radar detection measurement is an integration system, including a mechanical radar device, signal transmission device, and signal processing system. Two rotating radars are installed on the top of the blast furnace (see Figure 7). One of them only measures half of the burden surface profile from the centerline of the furnace to the periphery. They are symmetrically distributed in the top of the furnace and can detect the full material surface together.

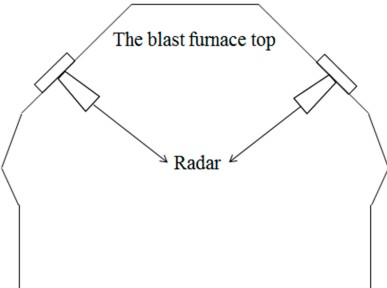

**Figure 7.** Schematic diagram of the radar installation in the top of the blast furnace.

After a new burden layer has been formed, the two radars are used to get the radar of the full surface. A combination of the mathematical model and radar data is divided into radar data processing and mathematical model calculation and is shown in Figure 8. After radar data processing, the burden profile function is obtained. The burden descent velocity is gained by two methods: The calculation descent function from the mathematical model and the fitting function of radar data. After the descent function, the burden profiles will present the layer structure. Then, we can evaluate the burden distribution through calculation of the ratio of ore to coke.

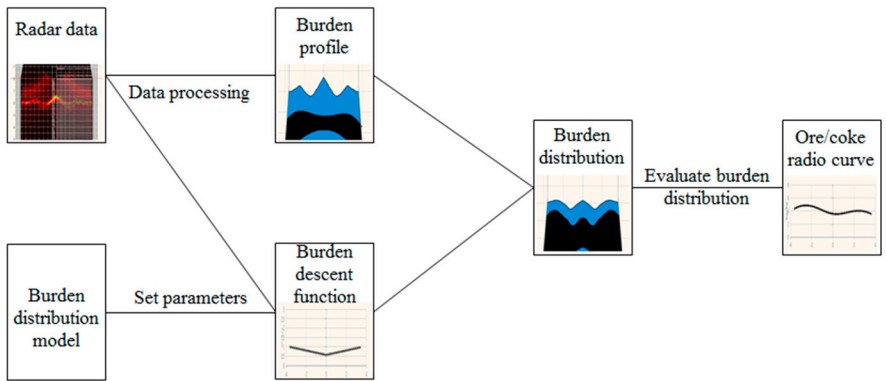

**Figure 8.** Procedure of the combination of the mathematical model and radar data in a software implementation.

### 2.2.1. Radar Data Collection

When a rotating radar works, it rotates around the axis of itself to detect the radial data of the burden profile. The radial data includes the height and the radius of the burden surface. Dust, chute shield, and airflow in the throat all interfere with the radar data, producing noise and making the measurement values deviate from the real ones.

### 2.2.2. Processing of Radar Data

Radar data were collected from a blast furnace of Nanjing Steel. In order to ensure the accuracy and authenticity of the data, the K nearest neighbor algorithm was firstly employed to remove the noise. Then, the Delaunay triangulation algorithm was used to realize the visualization of the 3D surface. The burden profile takes the average values of multiple coordinates extracted from the data. After the above processing, 40 group points were selected as the coordinates of the burden profile. The intersection of the furnace center line and the zero line of the stock line was defined as the origin of the coordinates. These 40 group points were converted into two-dimensional coordinates of the burden profiles.

The 40 groups of radar data were treated and stored in a database, which included the radar data time, burden profile coordinates (x, y), and material type as shown in Figure 9.

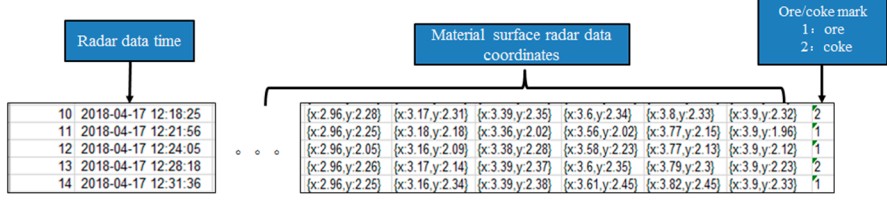

**Figure 9.** Radar data structure in the database.

From these 40 group of radar data, burden profiles were extracted and illustrated, as depicted in Figure 10. The radar data are independent points and discontinuous in the radial direction (red points

in Figure 10). Therefore, a polyfit regression method was used to find the most consistent curve for these radar data. This not only retains the characteristics of the radar data but also makes the burden profile look continuous and smooth.

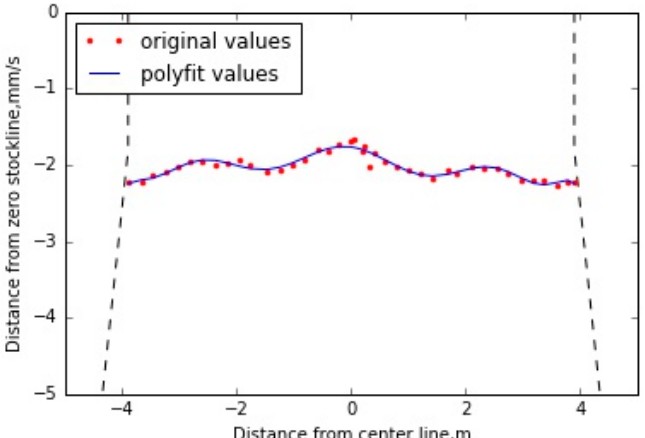

**Figure 10.** Radar data calculation of the burden profile function.

When the burden profile function is known, it is modified by the burden descent function. There are two methods to obtain the descent function: The one from the mathematical model (Equations (19)–(21)) and the other from the radar data fitting method.

Considering the difference between these two measurements, the descent velocity function can be calculated by:

$$V_d = \frac{f(r)_n - f(r)_{n-1}}{t_n - t_{n-1}}, \tag{24}$$

where $f(r)_n$ and $f(r)_{n-1}$ are the nth burden distribution and the (n − 1)th one after the descent, respectively. $t_n$ and $t_{n-1}$ define the charging times of the nth and (n − 1)th burden distribution, respectively.

A polyfit regression method was used to obtain the descent functions show in Figure 11. The distribution of the descent function is symmetrical at the centerline of the furnace (x = 0). The red dots in Figure 11 and blue curve express the radar data and the fitted curve, respectively.

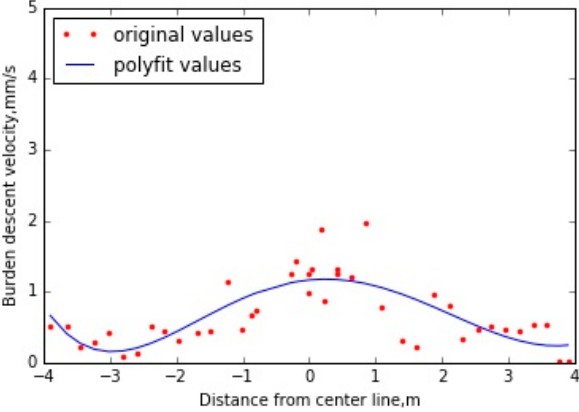

**Figure 11.** Method of the burden descent velocity from radar data.

2.2.3. Burden Distribution Calculation

From the above, the function of the burden profile and the function of the descent velocity were obtained. Then, the type of the charging material is identified by comparing the last batch with the

previous one. If the type is different, radar data are used to calculate the burden profile function. Otherwise, it is used to calculate the descent velocity and to modify the last material layer. After many layer calculations, the eburden forms a structure layer by layer as shown in Figure 12. Then, the coke to ore ratio curve of the burden can be calculated for the last two batches.

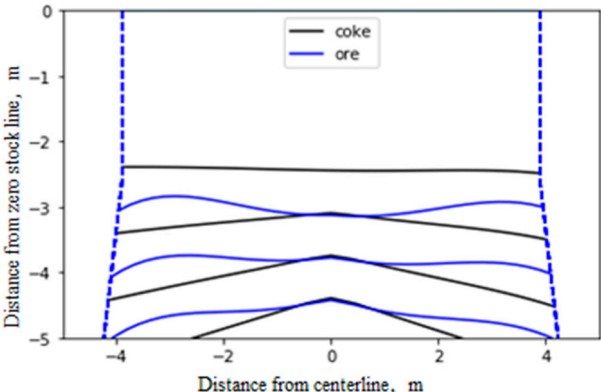

**Figure 12.** Burden distribution with layer by layer.

### 2.3. Parameters of Blast Furnace and Assumptions of Calculation

In order to combine the mathematical model with the radar data for a blast furnace, the relevant parameters of the charging system are listed in Tables 1–3.

**Table 1.** Parameters of the bell-less top blast furnace.

| Property | Value |
|---|---|
| Throat diameter (mm) | 8300 |
| Throat height (mm) | 2600 |
| Shaft angle (°) | 84.15 |
| $D_c$ [1] (mm) | 4010 |
| $D_b$ [2] (mm) | 1030 |
| $D_z$ [3] (mm) | 4601 |

[1] Distance from throttle to chute suspension point. [2] Distance from the chute suspension point to the chute bottom plate. [3] Distance from the chute suspension point to the zero line.

**Table 2.** Parameters of rotating chute.

| Property | Value |
|---|---|
| The length of the chute (mm) | 3890 |
| Chute speed (r·s$^{-1}$) | 0.133 |
| Coefficient of coke friction | 0.758 |
| Coefficient of ore friction | 0.638 |
| Velocity attenuation coefficient of coke | 0.70 |
| Velocity attenuation coefficient of ore | 0.71 |

**Table 3.** Physical parameters of raw material.

| Property | Ore | Coke |
|---|---|---|
| Bulk density (kg/m$^3$) | 1800 | 550 |
| Repose angle (°) | 31.5 | 32.5 |

Considering the influence of the charging parameters, burden distribution properties, and practical experience of blast furnace operators, the mathematical model combining the radar data was derived on the basis of the following assumptions [5,23]:

(1) Velocity of particles after collision with chute can be described by an attenuation factor without bouncing of particles in the chute.

(2) The chute rotates around the centerline of the blast furnace at any inclination angles with the revolution speed of 8 ring/min.

(3) There is no size distribution of particles in the raw material. The drag force of particles in the air after the chute and Coriolis force can be ignored.

(4) Burden is distributed in three dimensions, is uniform around the circumference, and is symmetrical around the centerline of the blast furnace.

(5) Burden keeps an alternate layer structure during the descent.

## 3. Application of the Combined Model and Results

### 3.1. Mathematical Model Test

The trajectory of the burden flow model was used to calculate the material flow path at different inclination angles of the chute as shown in Figure 13. With the increase of the inclination angle of the chute, the trajectory moves to the periphery. The drop point for a large inclination angle of the chute is farther away from the center line of the blast furnace. When the inclination angle of the chute is less than 15°, the chute dumps the materials directly to the center of the furnace (called center-charged burden), which cannot be observed in Figure 13. Figure 14 shows the radial velocity of the burden descent calculated by the burden decent velocity model (Equations (19)–(21)). The descent velocity increases with the increase of the distance from the centerline.

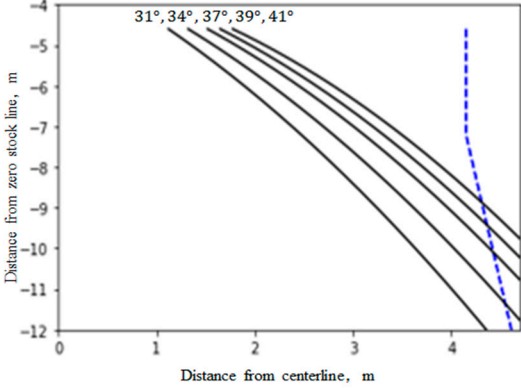

**Figure 13.** Main trajectories of coke flow.

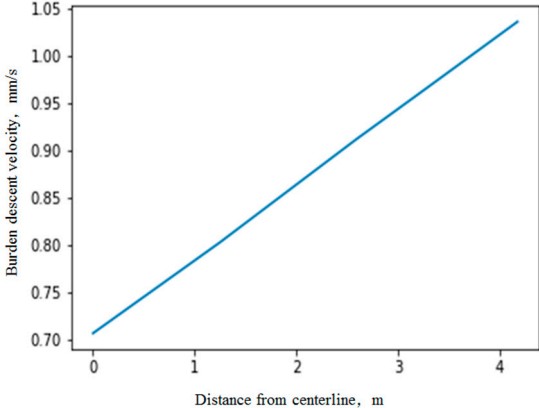

**Figure 14.** Burden descent velocity from the mathematical model.

Based on the parameters of Tables 1–3 and charge matrix of Table 4, the burden distribution of a multi-ring charging program was calculated by the burden profile model. The results are shown

in Figure 15. Blue, green, and red lines express the initial material surface, and the ore and coke surface, respectively. An ore profile with a single ring is shown in Figure 15a. After the full burden matrix, the burden distribution of a batch of ore and coke can be calculated as shown in Figure 15b. The structure of multi-batch burden layers (two coke and two ore layers) was calculated by iterative calculation, as shown in Figure 15c. From the latter, the same type of material has a similar shape and the apexes of the burden profile move toward the periphery a little during the descent due to the effect of the shaft angle of the furnace.

**Table 4.** Charging matrix of a blast furnace.

| Chute Angle (°) | 46 | 44 | 41.5 | 39 | 36.5 | 15 |
|---|---|---|---|---|---|---|
| Coke charging ring number | | 2 | 2 | 2 | 2 | 2 |
| Ore charging ring number | 3 | 3 | 2 | 2 | | |

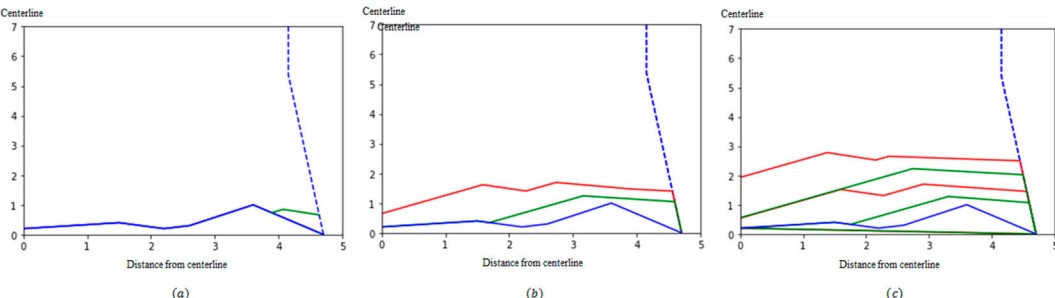

**Figure 15.** Burden distribution by a multi-ring charging program based on Tables 1–4: (**a**) 1st ring of ore; (**b**) a full burden distribution of an ore and a coke layer with multi-ring; (**c**) 2 coke and 2 ore layers. Blue line: initial material surface. Blue dash-dotted line: wall. Green line: ore layer. Red line: coke layer.

In order to test the mathematical model, a number of cases are listed in Table 5. A comparison of the burden distributions for different charging matrixes is shown in Figure 16. Figure 16a shows burden profiles with central coke when the inclination angle of the chute is 12° and the number of coke rings is 5. Figure 16b shows the burden profile with exactly the same parameters as in Figure 16a but for four coke rings. Comparing Figure 16a,b, only one ring of coke moves to an angle of 27°, which means the central coke becomes thinner and the thickness of the coke layer at an angle of 27° becomes bigger. With another ring of coke moving to 27° (Figure 16c), the thickness of the central coke becomes much thinner and the layer becomes much thicker than in Figure 16a. Figure 16d shows the burden distribution without the central coke and three rings moved to an angle of 20°. Compared to Figure 16c, the central coke has disappeared and the layer thickness at an angle of 20° becomes bigger than in Figure 16d. a comparison of Figure 16d,e shows that only a ring of coke moves from an angle of 20° to 27°. Therefore, the only difference between them is that the coke thicknesses at these two angles are somewhat different. Figure 16f shows the burden distribution with one ring less of coke at an angle of 27° compared to (e). Comparing Figure 16f,g, the burden distribution without two rings at an angle of 20° in the latter yields a thin layer at an angle of 20°. In Figure 16h, every inclination angle of the chute decreased by 1° except 27° for coke. Therefore, the ore layers move toward the center. Figure 17 shows the effect of the ore batch on the burden distribution (a. with ore batch of 63 t and b. with ore batch of 55 t). When the ore batch decreased from (a) to (b), the ore layer thickness became smaller.

**Table 5.** Charging programs with different matrixes to test the mathematical model.

| Figure No. | Inclination Angle of the Chute (°) | 41 | 39 | 37 | 34 | 31 | 27 | 20 | 12 |
|---|---|---|---|---|---|---|---|---|---|
| a | Ore | 2 | 3 | 3 | 2 | 1 | | | |
| | Coke | 3 | 3 | 3 | 2 | 2 | | | 5 |
| b | Ore | 2 | 3 | 3 | 2 | 1 | | | |
| | Coke | 3 | 3 | 3 | 2 | 2 | 1 | | 4 |
| c | Ore | 2 | 3 | 3 | 2 | 1 | | | |
| | Coke | 3 | 3 | 3 | 2 | 2 | 2 | | 3 |
| d | Ore | 2 | 3 | 3 | 2 | | | | |
| | Coke | 3 | 3 | 3 | 2 | | 2 | 3 | |
| e | Ore | 2 | 3 | 3 | 2 | 1 | | | |
| | Coke | 3 | 3 | 3 | 2 | 2 | 3 | 2 | |
| f | Ore | 2 | 3 | 3 | 2 | 1 | | | |
| | Coke | 3 | 3 | 3 | 2 | 2 | 2 | 2 | |
| g | Ore | 2 | 3 | 3 | 2 | 1 | | | |
| | Coke | 3 | 3 | 3 | 2 | 2 | 2 | | |
| | Inclination angle of the Chute (°) | 40 | 38 | 36 | 33 | 30 | 27 | | |
| h | Ore | 2 | 3 | 3 | 2 | 1 | | | |
| | Coke | 3 | 3 | 3 | 2 | 2 | 2 | | |

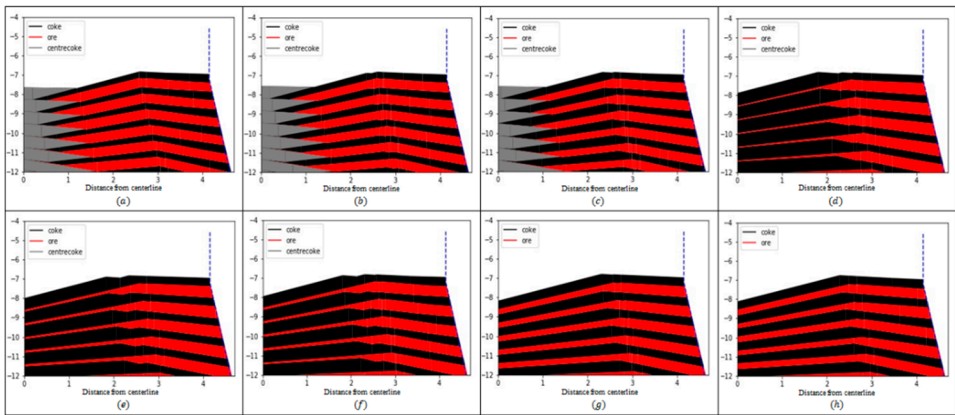

**Figure 16.** (**a–h**) are the comparison of burden distributions corresponding to the charging matrix of (**a–h**) in Table 5.

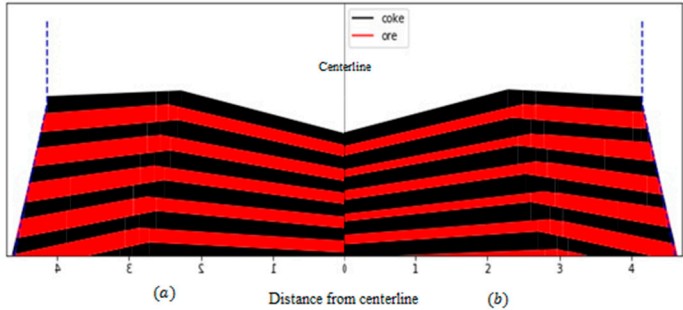

**Figure 17.** Comparison of the burden distribution with different ore batches: (**a**) Ore batch of 63 t; (**b**) Ore batch of 55 t.

Figures 16 and 17 shows the test of the sensitivity of the mathematical model. When the number of burden rings increased, the corresponding thickness of the burden layer increased. When the inclination angle of the chute changed, the structure of the burden layers changed accordingly. According to the

conservation of volume, the burden distribution changes with the change of the burden batch. In short, changes in the inclination angle of the chute, ring of charging, and ore batch will cause corresponding changes of the layers predicted by the mathematical model.

The burden distribution with the charging matrix of Table 5a is shown in Figure 18a. The ratio of ore to coke (Equation (23)) is defined by the last two layers and is shown in Figure 18b. The ratio is zero at the center due to the central-coke layers and has a highest value at r = 1.3 m at the inclination angle of the chute of 31°.

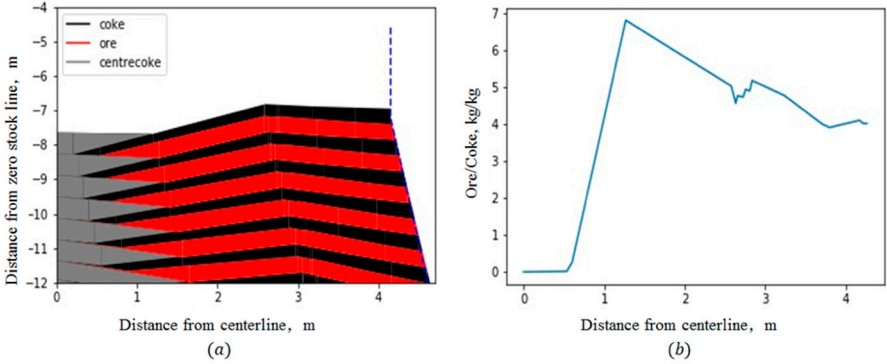

**Figure 18.** (**a**) Burden distribution; (**b**) ratio of ore to coke from (**a**) case.

### 3.2. Combination of the Mathematical Model and Radar Data

Radar data can work together with the mathematical model to support and guide the operation of blast furnace charging. Therefore, it is necessary to compile them into a visual interface. Based on the charging parameters and radar data from a plant in East China, a 2D simulation software of blast furnace charging was developed. The model was programed in Python and the visualization code was provided by JavaScript to give a friendly web interface.

A cloud map drawn by radar scanning data is shown in Figure 19a. Radar scanning data includes many points in an interval. After noise removal and feature extraction in the interval, 20 relatively stable points (green curve) were obtained. These points were used to calculate the burden distribution by the burden distribution model. The calculated burden layers' structure is shown in Figure 19b.

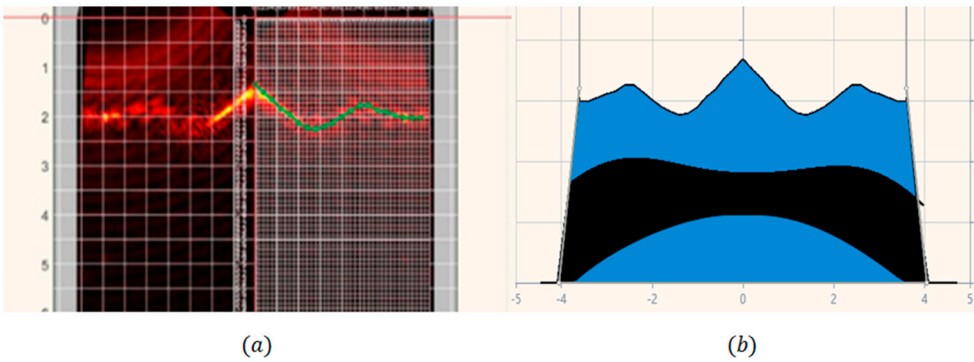

**Figure 19.** Burden profile: (**a**) Burden profile from radar data; (**b**) Burden profile from the mathematical model combined with radar data.

In order to display the radar data and the results of the mathematical model, we designed a user-friendly operator interface. Its main functions are shown in Figure 20. After logging into the software (Figure 20a), the left column is the function menu of the software. A column is designed to enter the parameter settings of the furnace in Figure 20b and includes blast furnace parameters, chute parameters, raw material properties parameters, charging matrix, and so on. The burden distribution display is the main page of the software and is drawn out by Echarts after radar data

processing in Figure 20c. After noise removal and feature extraction, the radar data combined with the mathematical model illustrate the structure of the material layers. The burden descent velocity and ratio of ore to coke are also drawn in the main interface. In addition, the system parameters of the radar are also monitored, such as the chip temperature, nitrogen temperature and nitrogen flow rate, valve pressure in radar system, and liquid flow rate.

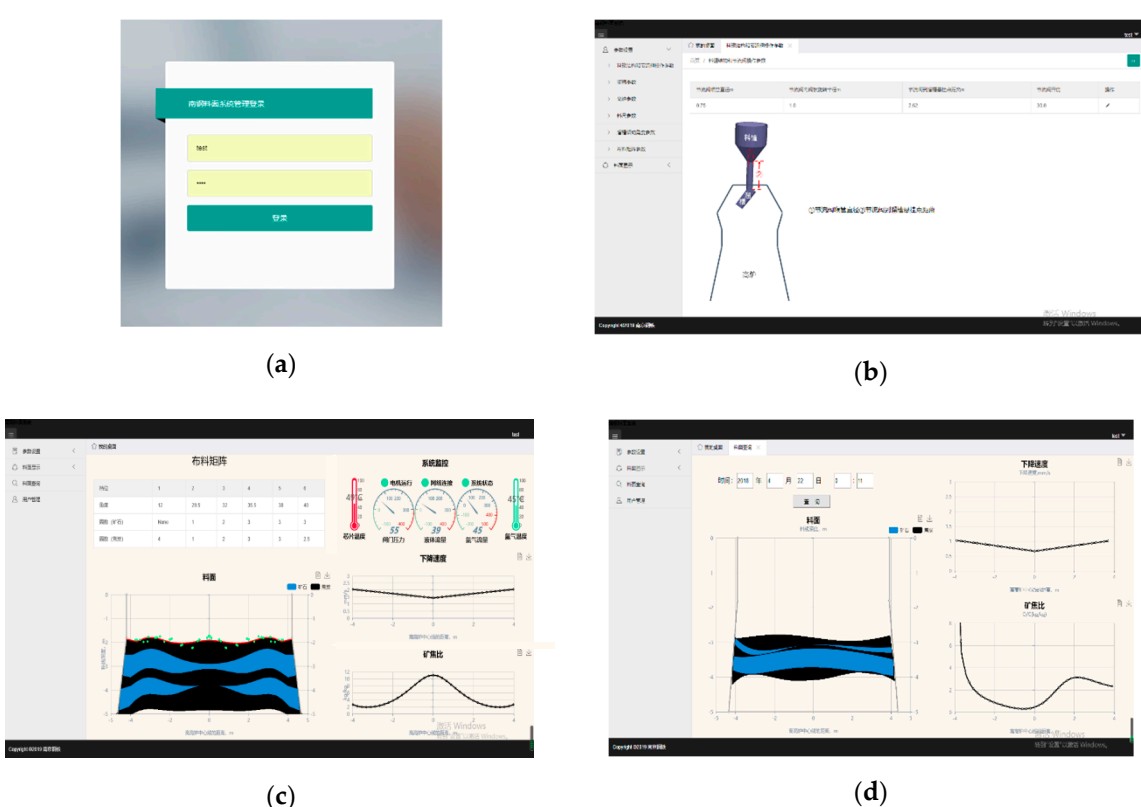

(**a**)

(**b**)

(**c**)

(**d**)

**Figure 20.** Visualization of our software by the combination of the mathematical model and radar data: (**a**) Software login interface; (**b**) Parameter setting interface; (**c**) Burden distribution display and radar data monitor; (**d**) User management interface.

In order to guarantee the security of the data and analyze the data of different users and different furnaces, a user management system was designed for user administration, as shown in Figure 20d.

## 4. Conclusions

The charging of the blast furnace directly affects the burden distribution in the throat, which influences the gas distribution in the shaft of the furnace. Adjusting the charging can improve the distribution of gas flow, increase the gas utilization efficiency, reduce energy consumption, and prolong the life of the furnace.

In this paper, with the help of computer technology, a mathematical model of the charging system was developed, composed of the trajectory of burden flow, burden profile, burden layer structure, and burden distribution evaluation. Serial cases were used to test the mathematical model's sensitivity. After noise removal and feature extraction, radar data of the burden profile was combined with the mathematical model to improve the accuracy of the model. A 2D view was obtained by combining the mathematical model and the radar data to visualize the charging, burden distribution, radar data, mathematical model, and relative equipment state. Based on the data from a blast furnace, the software was found to be more consistent than the present state-of-the-art tools used at the plant and ran smoothly to help the operation of the furnace.

**Author Contributions:** Conceptualization, M.L. and Y.Y.; methodology, M.L., H.W. and Y.Y.; software, M.L.; validation, Y.Y.; formal analysis, H.W. and Y.Y.; investigation, M.L. and Y.Y.; resources, G.X. and Y.Y.; data curation, Y.G. and G.X.; writing—original draft preparation, M.L. and Y.G.; writing—review and editing, H.W. and Y.Y.; visualization, M.L.; supervision, G.X. and Y.Y.; project administration, G.X. and Y.Y.; funding acquisition, Y.Y. All authors have read and agree to the published version of the manuscript.

**Funding:** We gratefully acknowledge financial support from The Program for Professor of Special Appointment (Eastern Scholar) at Shanghai Institutions of Higher Learning (No.TP2015039), National Natural Science Foundation of China (No.51974182), National 111 project, Grant/Award No.17002 and Jinheng information technology in Nanjing.

**Conflicts of Interest:** The authors declare no conflict of interest.

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
