# Peer review of "A Mathematical Model Combined with Radar Data for Bell-Less Charging of a Blast Furnace"

_processes, doi:10.3390/pr8020239_

Round 1

Reviewer 1 Report

The authors combined radar measurements and a mathematical model of
blast furnace charging  to control gas distribution and improve energy efficiency of the process.

The paper is well written, data supports the conclusions about the hypothesis.

In my opinion in lines 231-233 the authors should extensively describe how the K nearest neighbor algorithm has been used to remove the noise points in radar data and how Delaunay algorithm was used to
triangulate the point cloud and compress the feature extraction to 2D burden profile.

It is not so clear the discussion about the test of the sensitivity of the mathematical model used in my opinion the authors should expand lines 327-328 exhaustively to supports the conclusions

Reviewer 2 Report

The authors have conducted mathematical modelling of bell-less charging of blast furnace. The following comments should be addressed before manuscript is accepted for publication:

The authors have specified two models - charging and evaluation. In later sections, they just specified "Model". Which model are they referring to? What is the specifics of the model? Which software was used for coding and what are the assumptions. These data seem to be missing. The authors should conduct detailed literature review. Several important references are missing such as: Metals 20177(3), 83.
